# Fibrolytic rumen bacteria of camel and sheep and their applications in the bioconversion of barley straw to soluble sugars for biofuel production

Alaa Emara Rabee[1]*, Amr A. Sayed Alahl[2], Mebarek Lamara[3], Suzanne L. Ishaq[4]

1 Animal and Poultry Nutrition Department, Desert Research Center, Cairo, Egypt, 2 Renewable Energy Department, Desert Research Center, Cairo, Egypt, 3 Forest Research Institute, University of Quebec in Abitibi-Temiscamingue, Rouyn-Noranda, Canada, 4 School of Food and Agriculture, University of Maine, Orono, Maine, United States of America

* rabee_a_m@yahoo.com

**Data Availability Statement:** The data used in this study are available in the NCBI SRA Database at the accession number PRJNA743427 (https://www.ncbi.nlm.nih.gov/bioproject/PRJNA743427/).

## Abstract

Lignocellulosic biomass such as barley straw is a renewable and sustainable alternative to traditional feeds and could be used as bioenergy sources; however, low hydrolysis rate reduces the fermentation efficiency. Understanding the degradation and colonization of barley straw by rumen bacteria is the key step to improve the utilization of barley straw in animal feeding or biofuel production. This study evaluated the hydrolysis of barley straw as a result of the inoculation by rumen fluid of camel and sheep. Ground barley straw was incubated anaerobically with rumen inocula from three fistulated camels (FC) and three fistulated sheep (FR) for a period of 72 h. The source of rumen inoculum did not affect the disappearance of dry matter (DMD), neutral detergent fiber (NDFD). Group FR showed higher production of glucose, xylose, and gas; while higher ethanol production was associated with cellulosic hydrolysates obtained from FC group. The diversity and structure of bacterial communities attached to barley straw was investigated by Illumina Mi-Seq sequencing of V4-V5 region of 16S rRNA genes. The bacterial community was dominated by phylum Firmicutes and Bacteroidetes. The dominant genera were RC9_gut_group, *Ruminococcus*, *Saccharofermentans*, *Butyrivibrio*, *Succiniclasticum*, *Selenomonas*, and *Streptococcus*, indicating the important role of these genera in lignocellulose fermentation in the rumen. Group FR showed higher RC9_gut_group and group FC revealed higher *Ruminococcus*, *Saccharofermentans*, and *Butyrivibrio*. Higher enzymes activities (cellulase and xylanase) were associated with group FC. Thus, bacterial communities in camel and sheep have a great potential to improve the utilization lignocellulosic material in animal feeding and the production of biofuel and enzymes.

**Funding:** This project was supported by the USDA National Institute of Food and Agriculture, Hatch (or McIntire-Stennis, Animal Health, etc.) Project number ME0-22102 through the Maine Agricultural & Forest Experiment Station. Maine Agricultural and Forest Experiment Publication Number 3833. The funders had no role in study design, data collection and analysis, decision to publish, or preparation of the manuscript.

**Competing interests:** The authors have declared that no competing interests exist.

# 1. Introduction

The fermentation in the rumen depends mainly on the symbiotic relationships between complex networks of rumen microorganisms [1]. Bacteria are the primary taxa responsible for the fiber degradation in the rumen due to their large population and high activities [2]. Rumen archaea represent a small proportion of rumen microbiota; however, they achieve an important role by preventing the accumulation of gases in the rumen by reducing carbon dioxide using hydrogen to produce methane [3]. Methane contributes to global warming and represents a loss in dietary energy [4], and while they are important hydrogen sink in the rumen, there is interest in circumventing methane production in both ruminants and biodigester systems.

Ruminant animals in desert and tropical environments have been adapted to live in harsh conditions such as scarcity of water [5]. Moreover, they can utilize poor-quality fodder plants that are mostly avoided by other herbivores [6]. Barki sheep is the main breed in the desert of arid countries in North Africa and Middle East due to its adaptability to desert harsh conditions; however, it is lacking the appropriate feeding to increase their productive performance [7]. Dromedary camel has adapted to survive in harsh desert conditions by numerous unique characteristics and physiological mechanisms [8]. Therefore, camel and Barki sheep are preferred animals in the production system in the desert areas [7, 9]. Despite the economic importance of Bakri sheep and camel, the rumen microbiome has not received attention in comparison to other domesticated ruminants. In addition, the host-specific and diet-specific nature of microbial communities requires that investigation be conducted on these animals under current and future dietary management systems.

Barley is principal cereal crop in desert regions due to its adaptability to drought and soil salinity [10, 11]. In addition, it provides large quantities of lignocellulolsic-straw that is used in the feeding of desert ruminants such as camel, sheep, and goat to reduce the cost of feeding [12, 13]. The degradation of plant biomass in the rumen relies primarily on the colonization of rumen microbiota; therefore, understanding the plant-microbe interaction offers the possibilities to improve the fermentation efficiency to increase animal productivity [14]. Several studies were conducted to get insight into the fiber-attached bacteria in different ruminant species. Huws et al. (2016) [15] investigated the colonization of perennial ryegrass by rumen bacteria and found that the bacterial community was dominated by *Prevotella*, *Butyrivibrio*, *Fibrobacter*, *Pseudobutyrivibrio*, and *Selenomonas*. Similar findings were obtained by Du et al. (2019) [16] when steam-exploded wheat straw was inoculated by rumen fluid. Elliott et al. (2018) [17] investigated the colonization of different forage plants by rumen bacteria and observed that the chemistry of colonized plant is the main determiner of attached-bacterial community and the main colonizing bacteria were *Prevotella*, *Pseudobutyrivibrio*, *Ruminococcus*, *Olsenella*, *Butyrivibrio*, and *Anaeroplasma*.

Lignocellulosic biomass from plant residues is a suitable alternative to food crops in the production of second-generation bioethanol [18–20]. A huge amount of straw (dried stalks of grain plants) is generated globally every year; therefore straw is considered a renewable source of energy [18]. However, the biofuel production efficiency is limited due to slow hydrolysis of plant biomass [21, 22]. Therefore, different physical, chemical, and biological pretreatments are applied to improve the digestibility of lignocellulosic biomass [16, 22, 23].

Biological treatment involves using of cellulolytic enzymes or cellulolytic microorganisms [24]. The rumen is one of the most effective biological systems in the degradation of plant fiber that represent the major component of animal feed and the most abundant and renewable biomass on the earth [25]. Therefore, rumen fluid is a suitable source of mixed bacterial culture for effective cellulose degradation [26, 27], which could be applied in biogas and bioethanol production [22].

Furthermore, rumen microbiome is the key player in the variations in feed efficiency of ruminants [28]. Therefore, comparative studies that aim to understand the differences in rumen microbiome between ruminant species could lead to explain the fiber digestion and methane emission in efficient animals and identifying key microbes in rumen fermentation, which offer the possibility to regulate the rumen microbiota in less efficient animals towards more efficient microbial pathways and targeting of specific microorganisms to improve animal performance [8, 21, 29].

There is a lack of information about the lignocellulolytic bacterial communities in the rumen of camel and sheep and their ability to hydrolyze plant biomass to fermentable sugars. Therefore, this study aims to: (1) investigate the hydrolysis of barley straw by rumen bacteria of camel (*Camelus dromedarius*) and sheep (*Ovis aries*, Barki breed), (2) identify the lignocellulose-attached bacteria in the rumen of camel and sheep, (3) determine the effect of rumen inoculum source on gas, sugars, and enzyme productions from barley straw incubated with rumen culture of camel and sheep, and (4) demonstrate the possibility of ethanol production from hydrolysates generated from barley straw degradation by rumen bacteria.

## 2. Material and methods

### 2.1. Rumen samples

The experiment was conducted at Maryout Research, Desert Research Center, Alexandria, Egypt. Rumen contents were collected from three fistulated dromedary female camels (average body weight $455 \pm 5.5$ kg) and three fistulated barki rams (average body weight $50 \pm 1.7$ kg). Camels and sheep were fed diets consisting of 600 g/kg barley straw and 400g/kg concentrates mixture. The animals were housed individually in shaded pens with free access to drinking water. Concentrate feed mixture consisted of corn 60%, soybean meal 20%, wheat bran 24%, lime stone 2.5%, salt 1.5%, Sodium bicarbonate 0.5%, premix 0.3%, yeast 0.1%, and Antitoxins 0.1%. Rumen content was collected before morning feeding and strained through two layers of cheesecloth. The study was conducted under guidelines set by the Department of Animal and Poultry Production, Desert Research Center, Egypt. Moreover, the study was approved and all samples were collected in accordance with the guidelines of Institutional Animal Care and Use Committee, Faculty of Veterinary Medicine, University of Sadat City (Approval reference number: VUSC00008).

### 2.2. In vitro incubations

The barely straw was collected and dried at 65˚C, then was ground into 0.5 mm pieces. Then after, 0.5 g of ground material was weighted into individual 120 ml serum bottles. The growth medium that was used in this experiment is the modification of Medium 10 [30] was as follows (per 1000 ml distilled water): 2 g trypticase, 0.5 g yeast extract, 37 mL solution of $K_2HPO_4 \cdot 3H_2O$ (0.6 g in 100 mL distilled $H_2O$), 37 mL salt solution [0.16 g $CaCl_2 \cdot 2H_2O$, 0.6g $KH_2PO_4$, 1.2 g NaCl, 0.6 g $(NH_4)_2SO_4$, 0.25 g $MgSO_4 \cdot 7H_2O$ in 100 mL distilled $H_2O$], 1ml Hemin solution (1 g $L^{-1}$), 1mL Resazurin solution (1 g L-1), 50 ml solution of $Na_2CO_3$ (8g in 100 distilled $H_2O$), 1 g L-cysteine HCl, 200 ml clarified rumen fluid, 1 ml vitamin mix and 1ml trace mineral solution that were described by McSweeney et al. (2005) [31]. Also, clarified rumen fluid and anaerobic medium were prepared according to the protocol of McSweeney et al. (2005) [31]. Anaerobic medium (50 ml) was tubed into 120 ml-serum bottles under steam of $CO_2$. Strained rumen samples from each animal were kept under stream of $CO_2$; then 1 ml of every rumen sample was inoculated into its own serum bottle, and three replicate serum bottles were prepared for every rumen sample. Rumen bacteria were grown

anaerobically at 39˚C and at pH = 6.8 for 3 days. The growth was confirmed using microscopic examination and gas production.

## 2.3. Gas production

The cumulative total gas production after 24, 48, and 72 h of incubation was estimated in each bottle using graduated syringe displacement and the values were corrected for the blank value. The gas yield values are expressed in ml per 500 mg DM.

## 2.4. Dry matter disappearance

At the end of the 72-h fermentation, the contents of two serum bottles for each sample were filtered using nylon bags (8 cm wide 12 cm long, 42 mm aperture). Then after, the solid material was rinsed using distilled $H_2O$, oven-dried at 55˚C for 48 h, and weighed to determine the disappearance of dry matter and neutral detergent fiber (NDF). The pH was measured in fermentation liquid by digital pH meter (WPA CD70). The liquid samples were used to estimate lignocellulolytic enzymes (cellulase and xylanase), and reducing sugars (glucose, xylose), described below. Moreover, 50 ml of the liquid was used in an additional fermentation step to produce ethanol, described below. The content of the third bottle was filtered using nylon bags and the solid material was transferred to 50 ml tube for DNA isolation, described below.

## 2.5. Chemical analysis

Dry matter, crude protein (CP), and Neutral detergent fiber (NDF) were estimated in raw and fermented barely straw. Dry matter and crude protein (CP) were estimated according to AOAC, (1997) [32]. Neutral detergent fiber (NDF) was determined by the method of Van Soest et al. (1991) [33].

## 2.6. Lignocellulolytic enzymes and soluble sugars

Reducing sugars (xylose and glucose) and lignocellulolytic enzymes (cellulase and xylanase) were determined by the dinitro 3,5- salicylic acid (DNS) method [34, 35]. Xylanase was measured as endo-xylanase that was defined as the amount of enzyme that releases 1 μmol of xylose per ml in a minute. Cellulase was quantified as a unit of endo-β-1,4-glucanase that is defined as amount of enzyme that could hydrolyze filter paper and release 1 μmol of glucose within 1 a minute of reaction.

## 2.7. Bioethanol production

The liquid hydrolysates obtained from the growth of rumen bacteria on the barely straw for 72 h was inoculated with baker yeast and *Fusarium oxysporum* culture. The inoculated cultures were incubated at 30˚C for 48 hrs at pH 5.5 in an Erlenmeyer flask. After incubation, the fermented medium was centrifuged at 13,000 rpm for 10 min. The produced ethanol was determined in supernatant using the ethanol assay kits (Ben Biochemical Enterprise, Milano–Italy).

## 2.8. DNA isolation from solid material

The microbial cells' dissociation from incubated barley straw samples was conducted according to protocol described by Pope et al, (2010) [36]. Briefly, solid material was suspended in 15 mL dissociation solution (0.1% Tween 80, 1% methanol, and 1% tertiary butanol (vol/vol), pH 2). The mixture was vortexed for 1 min and centrifuged at 500 x g for 20s and then the supernatant was collected in sterile 50 ml tube. This step was repeated two more times and supernatants for each sample were collected and pooled. Microbial cell pellets were collected by

centrifugation at 12,000 × g for 5 min and subjected to DNA extraction by i-genomic Stool DNA Extraction Mini Kit (iNtRON Biotechnology, Inc.) according to the manufacturer's instructions. DNA was eluted in 50 μL elution buffer and DNA quality and quantity were verified through agar gel electrophoresis and Nanodrop spectrophotometer (Thermo Fisher Scientific, Madison, Wisconsin, USA). The V4-V5 region of the bacterial 16S ribosomal DNA gene was amplified using primers 515F (5′-GTGYCAGCMGCCGCGGTAA-3′) and 926R (5′-CCGY CAATTYMTTTRAGTTT-3′) [37]. PCR amplification was conducted in a thermal cycler under the following conditions: 94˚C for 3 min; 35 cycles of 94˚C for 45 s, 50˚C for 60 s, and 72˚C for 90s; and 72˚C for 10 min. PCR-products purification and preparation for sequencing were conducted according to protocol described by Comeau et al. (2017) [38]. The amplicons were then sequenced using Illumina MiSeq system in Integrated Microbiome Resource (Dalhousie University, Canada).

## 2.9. Real-time quantitative PCR (qPCR) of bacteria and archaea

Quantitative real-time PCR was conducted to determine the total bacterial and archaeal 16S rRNA copy number of straw-attached microbial communities. Bacterial standards were generated using dilutions of purified genomic DNA from *Prevotella sp*, *Ruminococcus albus*, and *Butyrivibrio hungatei* purchased from DSMZ (Braunschweig, Germany). Archaeal standards were generated using dilutions of purified genomic DNA from *Methanobrevibacter ruminantium*, and *Methanosphaera stadtmanae* purchased from DSMZ (Braunschweig, Germany). Dilution series of the standards ranging from $10^1$ to $10^6$ copies of the 16S rRNA gene were used. The qPCR was performed using Applied Biosystems StepOne system (Applied Biosystems, Foster City, USA). The bacterial specific primers F (5′-CGGCAACGAGCGCAACCC-3′) and R (5′-CCATTGTAGCACGTGTGTAGCC-3′) [39] and the archaeal specific primers Arch 1174–1195 F (5′-GAGGAAGGAGTGGACGACGGTA-3′) and Arch 1406–1389 R (5′-ACGGGC GGTGTGTGCAAG-3′) [40] were applied to amplify DNA samples and diluted standards. The 10-μL reaction consisted of 1 μL genomic DNA, 1 μL of each primer, and 7 μL SYBER Green qPCR- master mix (iNtRON Biotechnology, Inc.). The PCR conditions were as follows: 40 cycles of 95˚C for 15s, and 60˚C for 60s. The linear relationship between the threshold amplification (Ct) and the logarithm of 16S rDNA copy numbers of the standards was used to calculate the copy numbers of rumen bacteria and archaea per μL of DNA.

## 2.10. Bioinformatics analysis

The bioinformatics analyses of the paired-end (PE) Illumina raw sequences were processed in R (version 3.5.2) using DADA2 (version 1.11.3) [41]. Briefly, reads were denoised, dereplicated and filtered for chimeras to generate Amplicon Sequence Variants (ASVs) according to the recommended parameters in the DADA2 workflow. Taxonomic assignment of sequence variants was compared using the latest SILVA reference database [42]. The resulting ASV table was normalized and subsequently used to perform downstream analyses, including the computing of alpha and beta diversity metrics and taxonomic summaries.

## 2.11. Statistical analysis

The differences in DMD, NDFD, glucose and xylose yields, cellulase and xylanase activities, gas and ethanol productions, bacteria and archaea populations, and the relative abundances of bacterial phyla and genera were examined using unpaired T test at $P < 0.05$. Principal component analysis and Spearman's correlation were performed using the data of the relative abundance of dominant bacterial genera, DMD, NDF, sugars yield, enzymes activities, and gas production. The statistical analyses were performed using the SPSS v. 20.0 software package

[43] and PAST [44]. All the sequences were deposited to the sequence read archive (SRA) under the accession number: ##.

## 3. Results

### 3.1 Chemical composition and *In vitro* degradation of barely straw

The chemical composition of barely straw was as follow: 88% DM, 3.1% (Ether Extract) EE, 10.8% ash, 4.75% CP, 67% NDF. The results revealed that about 33.7% of DM and 28.5% of NDF were degraded within 72 h (Table 1). The source of inoculum did not affect DM degradability (DMD) significantly; however, higher DMD was observed with rumen inoculum of sheep (FR) compared to camel rumen inoculum (FC) (Table 1).

### 3.2 Lignocellulolytic enzymes, soluble sugars and ethanol production

The quantities of soluble sugars obtained from straw hydrolysis by different rumen inoculums are presented in Table 1. The results revealed that overall mean of glucose and xylose production were 9.27 and 0.13 µg/ml, respectively. The amount of sugars released was not affected by the animal donor; however, group FR exhibited higher glucose and xylose productions (Table 1). The overall mean of cellulase and xylanase production after 72 h was 0.51 and 0.42 IU/ml, respectively. Moreover, group FC displayed the highest enzyme activities. The overall ethanol yield was 0.9 mg/ml; and higher ethanol yield was obtained by FC group without significance difference (Table 1).

### 3.3 Gas production and microbial populations

Mean culture pH was 6.2 and group FC exhibited higher significant value than FR (Table 1). The total bacterial and archaeal copy numbers, and gas production at different fermentation

**Table 1. Effect of rumen inoculum source on dry matter and NDF disappearance (DMD, NDFD), and the production of sugars, enzymes, gas, and bioethanol from barley straw; and the population of bacteria and archaea.**

|  | Camel (FC) | SE | Sheep (FR) | SE | Overall mean | SEM | P-value |
|---|---|---|---|---|---|---|---|
| **Dry matter and NDF disappearance** | | | | | | | |
| **% DMD** | 33 | 0.72 | 34 | 0.53 | 33.7 | 0.46 | 0.515 |
| **%NDFD** | 28.5 | 1.6 | 28.5 | 0.34 | 28.5 | 0.74 | 0.916 |
| **Soluble sugars, Enzymes, and ethanol production** | | | | | | | |
| **Glucose µg/ml** | 8 | 1.4 | 10.6 | 0.78 | 9.27 | 0.92 | 0.181 |
| **Cellulase IU/ml** | 0.6 | 0.18 | 0.45 | 0.14 | 0.51 | 0.11 | 0.30 |
| **Xylanase IU/ml** | 0.6 | 0.06 | 0.24 | 0.1 | 0.42 | 0.1 | 0.245 |
| **Xylose µg/ml** | 0.11 | 0.02 | 0.15 | 0.04 | 0.13 | 0.02 | 0.277 |
| **Ethanol mg/ml** | 0.92 | 0.0295 | 0.88 | 0.004 | 0.9 | 0.01 | 0.302 |
| **16S rDNA -copies-Log 10/µL DNA for bacteria and archaea** | | | | | | | |
| **Bacteria** | 7.5 | 0.28 | 7.3 | 0.16 | 7.4 | 0.15 | 0.643 |
| **Archaea** | 4.7 | 0.3 | 5 | 0.06 | 4.8 | 0.16 | 0.558 |
| **Gas production (ml / 500 mg DM) at 24h, 48h, and 72h** | | | | | | | |
| **pH** | 6.2 | 0.016 | 6.1 | 0.01 | 6.2 | 0.03 | 0.008 |
| **GP-24 h** | 30.3 | 1.6 | 36 | 3.05 | 33.2 | 2 | 0.179 |
| **GP-48 h** | 34.7 | 1.3 | 44 | 1 | 39.3 | 2.2 | 0.005 |
| **GP-72 h** | 35.7 | 2.8 | 44.7 | 1.2 | 40.2 | 2.4 | 0.044 |

SE = Standard Error.

**Table 2. Alpha diversity indices of microbial community attached to barley straw as affected by rumen inoculum source.**

|  | Camel (FC) | SE | Sheep (FR) | SE | Overall mean | SEM | P-value |
|---|---|---|---|---|---|---|---|
| **Observed ASVs** | 1329.3 | 97.2 | 1823.3 | 264.9 | 1576.3 | 167.7 | 0.155 |
| **Chao1** | 1332.5 | 96.7 | 1825.8 | 266.04 | 1579.17 | 167.9 | 0.156 |
| **Shannon** | 6.8 | 0.082 | 6.96 | 0.25 | 6.88 | 0.12 | 0.586 |
| **InvSimpson** | 675.4 | 53.2 | 750.7 | 205.5 | 713.07 | 96.4 | 0.741 |

times, 24, 48. and 72h, and rumen inocula from camel and sheep are presented in Table 1. Compared to FC group, the FR group showed significantly higher gas production. The results of real-time PCR revealed that the means of bacterial and archaea population were 7.4±0.15 and 4.8±0.16 log 10 copy numbers/ μL, respectively. FC group showed greater bacterial population; and FR group showed higher archaeal population without significant differences.

## 3.4 Microbial diversity

A total of 214962 good quality sequence reads were generated from Illumina sequencing of six DNA samples with a mean of 35827±4032 (mean ± standard error (SE)) reads per sample. Total sequence reads in FC group was 81435 with a mean of 27145±2371 sequence reads. Total sequence reads in FR group was 133527 with a mean of 44509±565. Number of ASVs was more abundant in group FR than FC without significant difference (Table 2). Moreover, alpha diversity metrics, Chao1, and Shannon and Inverse Simpson followed the same trend (Table 2).

## 3.5 Analysis of bacterial community

The taxonomic analysis of bacterial communities revealed that straw-attached bacteria were classified into 5 phyla; Actinobacteria, Bacteroidetes, Firmicutes, Proteobacteria, and Synergistetes (Table 3). Phylum Bacteroidetes and Firmicutes make up more about 99% of bacterial community. Phylum Bacteroidetes accounted for 22.86% of bacterial community (Table 3). This phylum was assigned into three families; Muribaculaceae, Rikenellaceae, and Prevotellaceae. Furthermore, the members of Bacteroidetes were dominated by two genera, *RC9_gut_group*, and *Prevotella*. *RC9_gut_group* represented 22.2% of barely straw-associated bacteria and was more represented in FR group. Unlikely, genus *Prevotella* represented a small proportion (0.64%) and was higher in group FC (Table 4; Fig 1; S1 Table) without significance difference.

Phylum Firmicutes predominated the bacterial community (76.76%) and was higher in FC group (Table 3). On the family level, this phylum was classified into eight families; Ruminococcaceae, Lachnospiraceae, Veillonellaceae, Acidaminococcaceae, Family_XI, Streptococcaceae, Christensenellaceae, and Lactobacillaceae. On the genus level, this phylum was classified into 35 genera. Most of family Ruminococcaceae was assigned to genus *Ruminococcus*,

**Table 3. The relative abundance (%) of bacterial phyla colonized the barley straw incubated with rumen inoculum of camel and sheep.**

|  | Camel (FC) | SE | Sheep (FR) | SE | Overall mean | SEM | P-value |
|---|---|---|---|---|---|---|---|
| **Actinobacteria** |  |  | 0.28 |  |  |  |  |
| **Bacteroidetes** | 19.07 | 6.3 | 26.6 | 5.24 | 22.9 | 4.05 | 0.41 |
| **Firmicutes** | 80.89 | 6.37 | 72.64 | 5.18 | 76.8 | 4.1 | 0.372 |
| **Proteobacteria** | 0.11 | 0.005 | 0.3 | 0.1 | 0.2 | 0.04 | 0.001 |
| **Synergistetes** |  |  | 0.16 |  |  |  |  |

**Table 4. Relative abundances (%) of dominant bacterial families and genera attached to straw incubated with rumen inoculum of camel and sheep.**

| Family | Genus | Camel (FC) | SE | Sheep (FR) | SE | Overall mean | SEM | P-value |
|---|---|---|---|---|---|---|---|---|
| **Phylum: Actinobacteria** | | | | | | | | |
| **Atopobiaceae** | Olsenella | | | 0.28 | | | | |
| **Phylum: Bacteroidetes** | | | | | | | | |
| **Muribaculaceae** | | | | 0.08 | | | | |
| **Rikenellaceae** | RC9_gut_group | 18.28 | 6.1 | 26.09 | 5.2 | 22.2 | 3.99 | 0.386 |
| **Prevotellaceae** | Prevotella_1 | 0.79 | 0.25 | 0.48 | 0.2 | 0.64 | 0.16 | 0.333 |
| **Phylum: Firmicutes** | | | | | | | | |
| **Ruminococcaceae** | | 24.78 | 3.3 | 18.4 | 2.44 | 21.59 | 2.3 | 0.196 |
| **Ruminococcaceae** | Ruminococcus_1 | 14.8 | 3.2 | 9.95 | 1.34 | 12.4 | 1.9 | 0.234 |
| **Ruminococcaceae** | Saccharofermentans | 8.6 | 1.25 | 6.5 | 1.76 | 7.6 | 1.07 | 0.399 |
| **Ruminococcaceae** | NK4A214_group | 0.74 | 0.32 | 0.65 | 0.32 | 0.7 | 0.2 | 0.86 |
| **Ruminococcaceae** | UCG-004 | 0.23 | 0.16 | 0.3 | 0.03 | 0.27 | 0.07 | 0.667 |
| **Ruminococcaceae** | UCG-005 | 0.16 | 0.03 | 0.17 | 0.05 | 0.17 | 0.02 | 0.912 |
| **Lachnospiraceae** | | 30.4 | 3.2 | 37.1 | 5.3 | 33.7 | 3.12 | 0.343 |
| **Lachnospiraceae** | Pseudobutyrivibrio | 0.93 | 0.36 | 0.66 | 0.15 | 0.8 | 0.18 | 0.521 |
| **Lachnospiraceae** | Butyrivibrio_2 | 2.47 | 0.7 | 1.6 | 0.2 | 2.04 | 0.39 | 0.319 |
| **Lachnospiraceae** | probable_genus_10 | 6.09 | 1.39 | 20.4 | 8.5 | 13.26 | 5.02 | 0.172 |
| **Lachnospiraceae** | FCS020_group | 8.48 | 2.4 | 7.5 | 2.14 | 8.02 | 1.47 | 0.792 |
| **Lachnospiraceae** | AC2044_group | 3.54 | 0.13 | 2.66 | 0.55 | 3.1 | 0.32 | 0.198 |
| **Lachnospiraceae** | NK4A136_group | 3.59 | 0.97 | 1.36 | 0.06 | 2.48 | 0.66 | 0.149 |
| **Lachnospiraceae** | NA | 1.01 | 0.4 | 0.87 | 0.1 | 0.94 | 0.193 | 0.754 |
| **Lachnospiraceae** | FD2005 | 0.46 | 0.26 | 0.28 | 0.026 | 0.37 | 0.12 | 0.544 |
| **Lachnospiraceae** | Oribacterium | 2.3[a] | 0.38 | 0.82[b] | 0.22 | 1.57 | 0.39 | 0.027 |
| **Lachnospiraceae** | UCG-009 | 0.23[a] | 0.02 | 0.12[b] | 0.02 | 0.17 | 0.027 | 0.036 |
| **Lachnospiraceae** | Lachnoclostridium_1 | 0.14 | 0.01 | 0.19 | 0.05 | 0.17 | 0.02 | 0.433 |
| **Acidaminococcaceae** | Succiniclasticum | 15 | 4.4 | 6.2 | 0.44 | 10.6 | 2.79 | 0.12 |
| **Veillonellaceae** | | 3.67 | 2.081 | 4.6 | 2.34 | 4.13 | 1.42 | 0.781 |
| **Veillonellaceae** | Schwartzia | 0.32 | 0.03 | 0.34 | 0.055 | 0.3 | 0.03 | 0.764 |
| **Veillonellaceae** | Selenomonas_1 | 3.34 | 2.08 | 4.26 | 2.29 | 3.8 | 1.4 | 0.783 |
| **Family_XI** | | 0.2 | 0.02 | 0.49 | 0.13 | 0.35 | 0.08 | 0.113 |
| **Streptococcaceae** | Streptococcus | 1.97 | 1.09 | 0.6 | 0.19 | 1.29 | 0.58 | 0.291 |
| **Christensenellaceae** | R-7_group | 4.7 | 0.56 | 5.16 | 0.95 | 4.9 | 0.5 | 0.699 |
| **Phylum: Proteobacteria** | | | | | | | | |
| **Succinivibrionaceae** | Succinivibrio | 0.05 | | 0.22 | | | | |
| **Synergistetes** | Pyramidobacter | | | 0.16 | | | | |

*Saccharofermentans* that were higher in FC group (Table 4; Fig 1; S1 Table). Some genera within Ruminococcaceae were found in specific group such as Ruminococcaceae UCG-002, and Ruminococcaceae UCG-010 that were found in FR group. Furthermore, *Papillibacter* was observed only in FC group (Table 4).

Family Lachnospiraceae was the most predominant family in the Firmicutes phylum and represented 33.7% of sequence reads. This family showed higher relative abundance in FR group, and was dominated by *Butyrivibrio, Oribacterium,* and unclassified bacteria such as Lachnospiraceae FCS020_group, Lachnospiraceae AC2044_group, Lachnospiraceae NK4A136_group that were higher in FC group compared to FR with significant difference in *Oribacterium* (Table 4; Fig 1; S1 Table). Also, Lachnospiraceae probable_genus_10 make up 13.26% of microbial community and was higher in FR group. Family Acidaminococcaceae

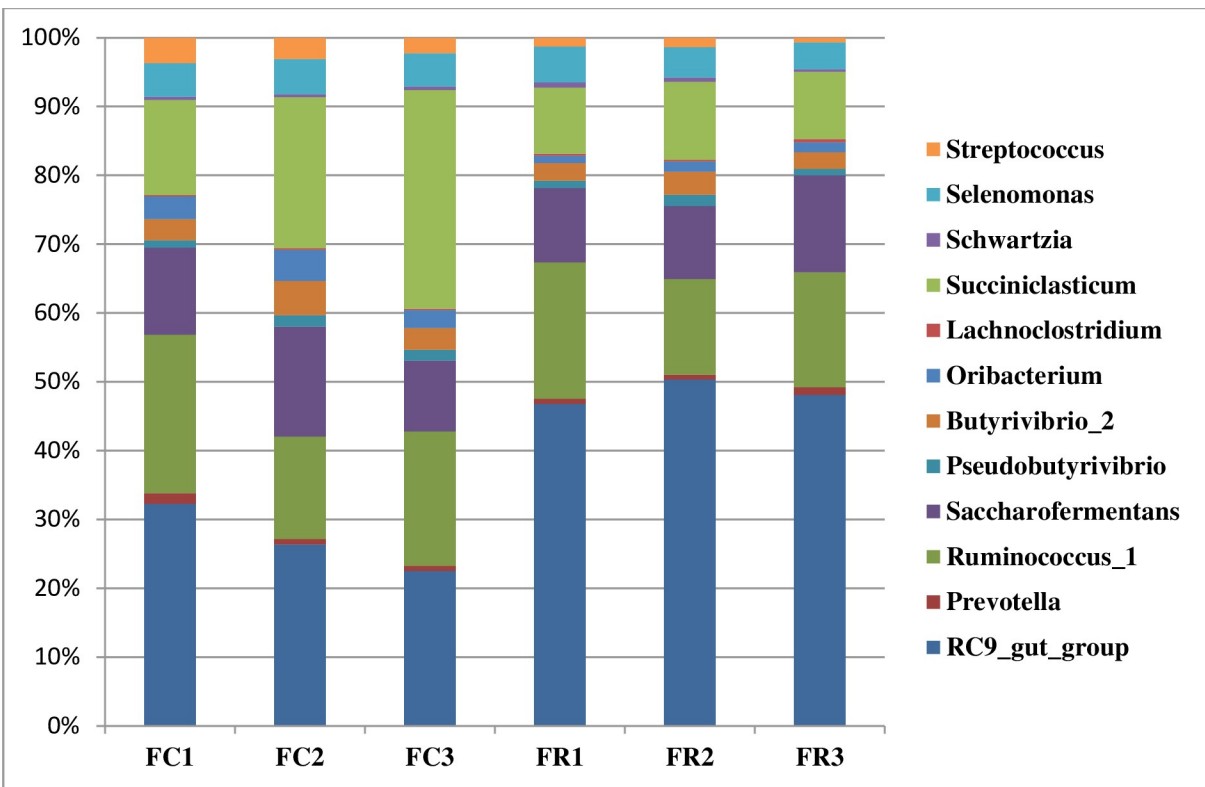

**Fig 1. The relative abundance of bacterial genera after 72h incubation.** The relative abundance of dominant bacterial genera attached to barley straw inoculated with camel rumen inocula (FC1-FC3) or sheep rumen inocula (FR1-FR3).

accounted for 10.6% of bacterial community and was higher in group FC compared to FR group (Table 4). The members of this family were assigned to *Succiniclasticum*. Most family Veillonellaceae were assigned to genus *Selenomonas* that was more prevalence in group FR. Family Streptococcaceae make up 1.29% and was further classified to genus *Streptococcus* that was higher in FC group. Family Christensenellaceae accounted for 4.9% and was dominated by R-7_group that was more abundant in FR group (Table 4; Fig 1; S1 Table).

Phylum Proteobacteria represented less than 0.5% of the bacterial community and was dominated by *Succinivibrio* that was higher in FR group. Additionally, Phylum Synergistetes and Actinobacteria were observed only in FR group and were predominated by genus *Pyramidobacter* and *Olsenella*, respectively (Tables 3 and 4; S1 Table).

### 3.6 Comparison between the rumen inocula of camel and sheep

Beta diversity of microbial community attached to barley straw was calculated using principal coordinate analysis (PCoA) based on Bray-Curtis dissimilarity (Fig 2). The results demonstrated that microbial communities of FC and FR groups were separated distinctly. Venn diagram was conducted to explain the distribution of bacteria between straw-colonizing bacterial communities (FC, FR) (Fig 3). The diagram showed that 66.7% of ASVs were shared between the bacterial groups. Moreover, Group FR revealed higher unique ASVs compared to FC group. Principal component analysis (PCA) separated the samples based on the relative abundance of bacteria, DMD, NDF, sugars yield, enzymes activities, and gas production; the differences between the samples were driven by the relative abundance of family Ruminococcaceae, genus RC9_gut_group, probable_genus_10, and *Selenomonas* (Fig 4).

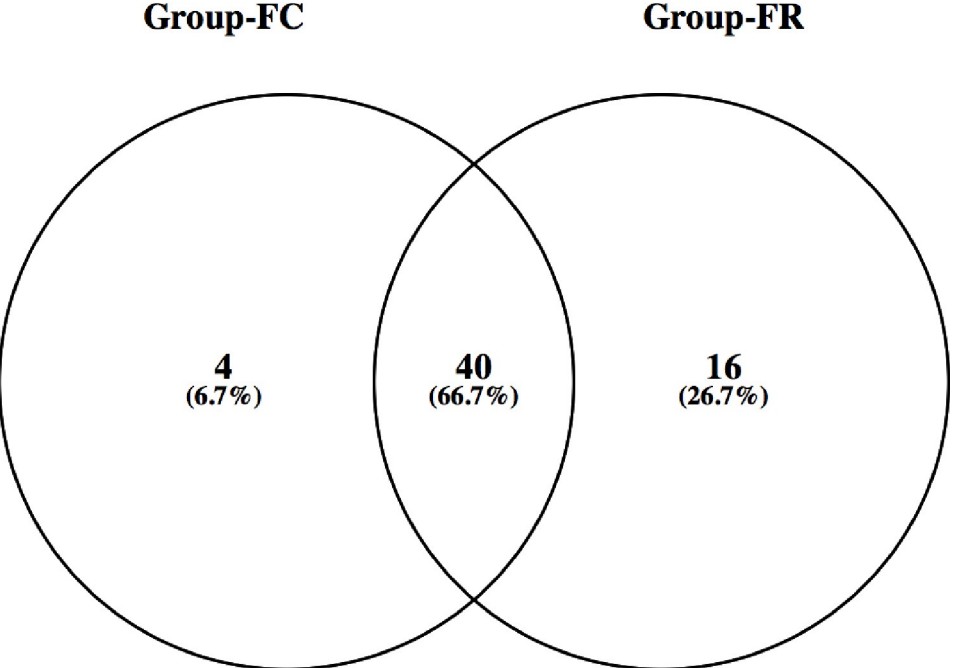

**Fig 2. Principal coordinate analysis (PCoA) of bacteria attached to barley straw based on Bray-Curtis dissimilarity.** Green circles for bacteria of camel rumen inoculum (FC). Blue triangles for bacteria of sheep rumen inoculum (FR).

### 3.7 Pearson correlation analysis

Pearson correlation analysis (Fig 5) showed some positive and negative correlations relationships. Gas production correlated positively with glucose, xylose and RC9_gut group and negatively with Xylanase, *Butyrivibrio*, and *Streptococcus*. DMD correlated positively with cellulase,

## Group-FC          Group-FR

**4**
**(6.7%)**

**40**
**(66.7%)**

**16**
**(26.7%)**

**Fig 3. Venn diagram of ASVs among the bacterial libraries.** The diagram illustrates number of unique ASVs for barely straw-colonizing bacteria originated from camel (FC), and sheep (FR); and the overlapping area represents the shared ASVs between FC and FR groups.

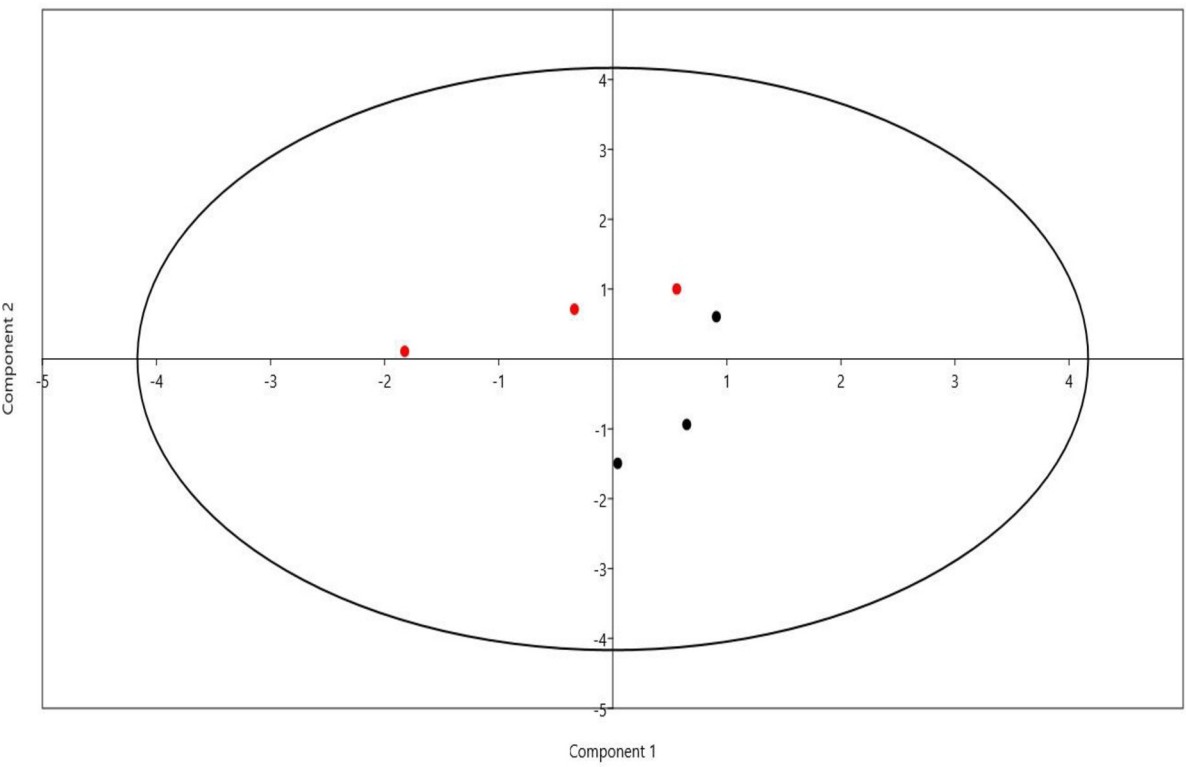

**Fig 4. Principal component analysis (PCA).** Principal component analysis was conducted based on the relative abundance of dominant bacterial genera, DMD, NDF, sugars yield, enzymes activities, and gas production. Black circles for rumen inoculum of camels (FC), and red circles for rumen inoculum of sheep (FR).

glucose, and RC9_gut group. NDFD correlated positively with cellulase, glucose, RC9_gut_group, *Prevotella*, and *Ruminococcus* and negatively with *Pseudobutyrivibrio*. Glucose correlated positively with RC9_gut_group and negatively with *Succiniclasticum* and *Streptococcus*. *Prevotella* correlated positively with *Ruminococcus*.

## 4. Discussion

The rumen of desert-dwelling and wild ruminants such as camel and sheep is considered an untapped source of lignocellulolytic enzymes and bacterial strains [1, 27]. Understanding the colonization of cellulolytic biomass by rumen bacteria is the cornerstone for developing bacterial mixed consortium to improve the deconstruction of cellulolytic biomass to be used in animal feeding or biofuel production [19, 25, 45]. Moreover, cellulolytic consortium from efficient animals could be transferred to less efficient animals to improve rumen fermentation and animal production [46]. Barley straw is an abundant lignocellulosic biomass worldwide and has higher fiber content and low protein content [47]. The current study aims to investigate the hydrolysis and colonization of barley straw by rumen inoculums from camel and sheep. The chemical composition of barely straw in our study was in the range indicated in previous studies [21, 48]. Viljoen et al. (2005) [48] reported that barely straw had higher NDF and CP and the lowest DM digestibility compared to wheat and oat straw. Therefore, it's necessary to improve the nutritive value of the barely straw to be used as roughage in animal feeding or feedstock in biofuel production.

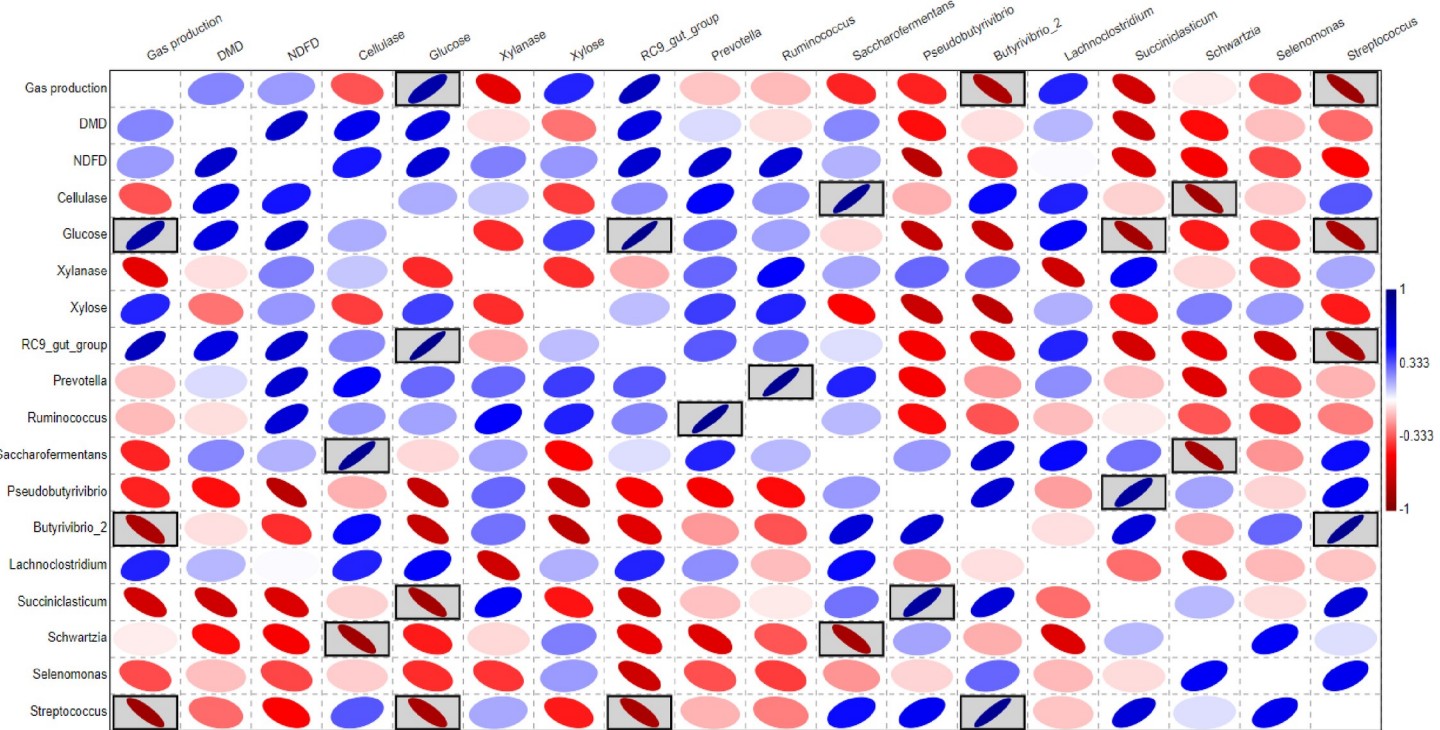

**Fig 5. Pearson correlation analysis; the analyses showed some positive and negative correlations.** For example, gas production correlated positively with glucose, xylose and RC9_gut group; and negatively with Xylanase, *Butyrivibrio*, and *Streptococcus*. DMD correlated positively with cellulase, glucose, and RC9_gut group. NDFD correlated positively with cellulase, glucose, RC9_gut_group, *Prevotella*, and *Ruminococcus* and negatively with *Pseudobutyrivibrio*. Glucose correlated positively with RC9_gut_group and negatively with *Succiniclasticum* and *Streptococcus*. *Prevotella* correlated positively with *Ruminococcus*.

## 4.1 Hydrolysis of barely straw, ethanol and gas production

The hydrolysis of lignocellulose is the primary step to converting plant biomass to biofuel [22]. The degradability of DM and NDF in this study (Table 1) in the range reported by previous studies [19, 29, 48]. Oss et al., (2016) [29] inoculated barley straw by different proportions of cattle: bison rumen inoculums and found no effect of rumen inoculum source on disappearance of DM and NDF, which supports our results. In contrast, He et al. (2019) [49] incubated different substrates, including barley straw with different combination of rumen inocula of cattle and bison and reported that rumen inoculum source affected the DMD and NDFD. Poszytek et al., (2016) [22] reported that DMD of maize silage was 30% after incubation for 72h with cellulolytic consortium that in the line with our findings.

The efficient degradation of lignocellulose requires synergistic work of different types of cellulases and xylanases that produced by several microbial strains [45]. Thus, understanding of the enzymology of plant cell wall hydrolysis is the key step to the sustainable utilization of cellulosic biomass [36]. Cellulase and xylanase production by rumen inocula (Table 1) from camel and sheep were higher than the production of bacterial cultures isolated from camel rumen by Srivastava et al. (2021) [50]. Moreover, cellulase production was similar to the production of some commercial bacterial strains such as *E. coli* [18] and cellulolytic bacterial consortium [22], which highlight the higher cellulolytic activities of rumen bacteria in sheep and camel. Poszytek et al. (2016) [22] observed the highest cellulase activity and glucose production at 72h, at which the cellulose was hydrolyzed effectively; this finding supports our results.

Xylose and glucose yields (Table 1) were slightly higher than study by Du et al. (2019) [16]. Moreover, the glucose content is similar to values obtained from wheat straw incubated with

cellulolytic consortium [22] or untreated wheat straw [20]. Higher xylose and glucose in FR groups is a result of efficient hydrolysis of hemicelluloses and cellulose, which highlights the potential of using rumen inoculum of Barki sheep in the bioconversion of lignocellulosic biomass compared to rumen inoculum of camel [16].

The amount of fermentable sugars is the main determiner of ethanol production from cellulosic biomass [18, 51]. Ethanol yield was similar to values obtained from wheat straw by Momayez et al., (2017) [20]. Mutreja et al. (2011) [18] used enzymatic and chemical treatments in the bioconversion of eight lignocellulosic substrates into ethanol, and found that ethanol yield ranged from 0.24 to 1.2 g/ L, which agrees with our findings.

The values of pH in the current study (Table 1) in the optimal range of microbial growth especially the cellulolytic bacteria [45, 52]. Lower pH value in FR group compared to FC might be attributed to higher production of organic acids [53], and this trend is supported by higher sugars yield that could be fermented to volatile fatty acids [16]. The availability of soluble sugars is attributed to higher DM degradation that improves the fermentation [16, 22]. Furthermore, the fermentation of cellulose and hemicellulose generates hydrogen, carbon dioxide, and soluble sugars that are involved in the methanogenesis by rumen archaea [3, 54], which might demonstrate the positive correlation between gas production (GP) and glucose and xylose (Fig 5). Subsequently, higher archaeal population and GP in group FR are explained. Previous study by Rabee et al. (2020) [8] indicated that fibrous diets stimulate the population of rumen archaea.

The values of gas production (Table 1) were similar to values obtained from wheat and barley straw and corn stover in previous studies [29, 55, 56]. He et al. (2019) [49] stated that rumen inoculum source affected GP from barley straw incubated with different combination of rumen inocula of cattle and bison, which support the significant difference in GP by rumen inocula from camel and sheep in our study.

## 4.2 Analysis of bacterial community

The majority of bacterial communities were assigned to phylum Firmicutes and Bacteroidetes, which agrees with previous studies conducted on lignocellulosic forages (Table 3) [17, 57–60]. Phylum Firmicutes dominated the bacterial community attached to barley straw (Table 3), which is consistent with Du et al. (2019) [16] and explains that this phylum is the main contributor in lignocellulose degradation. On the genus level, phylum Firmicutes was dominated by *Ruminococcus*, *Saccharofermentans*, *Butyrivibrio*, *Succiniclasticum*, *Selenomonas*, *Streptococcus*, this finding is supported by studies on different lignocellulosic plants such as wheat straw, switchgrass, and ryegrass [15,16, 57, 60] (Table 4; Fig 1). These genera play important roles in the degradation of plant polysaccharides [16]. Genus *Ruminococcus*, and *Butyrivibrio* that predominated phylum Firmicutes, degrade the hemicellulose, pectin, and cellulose present in the plant cell wall and produce several types of cellulolytic and hemicellulolytic enzymes [8, 45, 61]. Thus, the positive correlation between *Ruminococcus* and NDFD is explained and was also reported by Liu et al. (2016) [57] (Fig 5). Furthermore, genus *Succiniclasticum* was found in the rumen of grazing cow [62] and has the capability to degrade fiber and cellobiose [16], which could illustrate the positive correlation between *Succiniclasticum* and xylanase (Table 4; Fig 5). Therefore, higher enzyme activities in FC could be attributed to the abundance of *Ruminococcus*, *Butyrivibrio*, and *Succiniclasticum* (Tables 1 and 4; Fig 1). This finding also supported by Dai et al. (2015) [63] who indicated that most of cellulases and hemicellulases in rumen produced by *Ruminococcus*. Unclassified bacteria represented higher proportion of phylum Firmicutes (Table 4); these bacteria might have a role in the degradation of plant biomass [8, 19]. Cheng et al. (2017) [19] indicated that more studies on unidentified bacteria are

needed to improve the utilization of cellulosic biomass and using them in industrial applications [64].

Members of phylum Bacteroidetes degrade a wide range of substrates, including cellulose, pectin and soluble polysaccharides and unclassified Bacteroidetes are more specialized in lignocellulose degradation [65]. All the members of phylum Bacteroidetes were assigned to genus *Prevotella* and RC9_gut_group, and that agrees with other studies [8, 19] (Table 4). Genus *Prevotella* is fibrolytic bacteria which use different substrates in growth, including hemicellulose, pectin, proteins and peptides [8, 57], which could demonstrate the positive correlation between NDFD and genus *Prevotella*. Liu et al. (2016) [57] reported a positive correlation between protein content and the relative abundance of *Prevotella*, which could explain the lower abundance of this genus in the current study compared to other studies [8] as the barley straw have low protein content.

On the other hand, genus *Prevotella* utilizes the hydrogen produced from cellulose fermentation in propionic acid production, which highlights the negative correlation between gas production and *Prevotella* [57, 66] (Fig 5). The RC9_gut_group belongs to uncultured Bacteroidetes that have a potential role in lignocellulose digestion [65], which could illustrate the positive correlation between RC9_gut_group and NDFD, and glucose production. Qian et al. (2019) [62] reported that RC9 gut group and *Prevotella* were the most dominant genera in bacterial community colonizing the reeds and cottonseed hulls in the rumen of Tarim red deer. Consequently, higher glucose in FR group could be attributed to the abundance of RC9_gut_group (Tables 1 and 4). The prevalence of phylum Bacteroidetes in FR group highlights the potential of rumen culture from sheep to be used in the biological treatments. Moreover, Henderson et al. (2015) [54] reported that both of RC9 gut group and *Prevotella* dominated the rumen bacteria in several ruminant species.

Poszytek et al., (2016) [22] reported that the cellulolytic bacterial consortium achieve better cellulose hydrolysis than single isolates, which highlights the rumen cultures from camel and sheep as a source of bacterial consortia with industrial applications [45]. Our results revealed that the bacterial community colonizing barley straw have several non-cellulolytic bacteria such as *Lactobacillus*, *Streptococcus*, *Selenomonas*; these genera use the end products of other bacteria in their growth [8]. According to Lewis et al. (1988) [67] the presence of non-cellulolytic bacteria in the bacterial consortium is important to utilize soluble sugars that are produced from the hydrolysis of cellulose; this process could avoid the feedback inhibition of cellulose-degrading bacteria [45].

Animal species and the chemical composition of animal diet are the main determiners of the rumen microbiome [54]. The findings of this study showed that the proportions of solid-attached Ruminococcaceae and RC9_gut_group are higher in this study than their proportions in camel and sheep in previous studies [8, 68] as well as cow [57], buffalo [69], and deer [62]. Furthermore, the relative abundance of *Prevotella* was lower than in other ruminant animals [8, 57, 62, 68, 69]. More details about the comparison between this study and previous studies could be found in Supplementary figure (S1 Fig).

## 5. Conclusion

The study demonstrated the degradation and colonization of barley straw by rumen bacteria of camel and sheep. The results revealed changes in the diversity and relative abundance of bacteria attached to barely straw between rumen inoculum of camel and sheep. Higher DMD, sugars release and gas production was obtained when barley straw was inoculated with sheep rumen inoculum. Rumen inoculum of camel demonstrated higher cellulolytic bacteria and enzyme activities. Higher ethanol production was obtained from hydrolysate of camel rumen

inoculum. Rumen of sheep and camel are promising source of cellulolytic bacteria and enzymes with industrial applications.

## Supporting information

**S1 Table. Relative abundances bacterial families and genera attached to straw incubated with rumen inoculum of camel and sheep.**
(DOCX)

**S1 Fig. The relative abundances of main rumen bacterial groups in different animal species.** A comparison of the relative abundances of the main bacterial groups in camel in the current study (Camel-A) and a previous study (Camel-B), Sheep in the current study (Sheep-A) and a previous study (Sheep-B), cattle, buffalo, and deer.
(PDF)

## Acknowledgments

Many thank the staff of Maryout Research station, Desert Research center, Egypt.

## Author Contributions

**Conceptualization:** Alaa Emara Rabee, Amr A. Sayed Alahl, Mebarek Lamara, Suzanne L. Ishaq.

**Data curation:** Alaa Emara Rabee, Mebarek Lamara, Suzanne L. Ishaq.

**Formal analysis:** Alaa Emara Rabee, Amr A. Sayed Alahl.

**Funding acquisition:** Amr A. Sayed Alahl.

**Investigation:** Alaa Emara Rabee, Amr A. Sayed Alahl, Mebarek Lamara, Suzanne L. Ishaq.

**Methodology:** Alaa Emara Rabee.

**Project administration:** Alaa Emara Rabee.

**Resources:** Alaa Emara Rabee, Amr A. Sayed Alahl, Mebarek Lamara, Suzanne L. Ishaq.

**Software:** Mebarek Lamara, Suzanne L. Ishaq.

**Supervision:** Suzanne L. Ishaq.

**Validation:** Alaa Emara Rabee, Mebarek Lamara, Suzanne L. Ishaq.

**Visualization:** Alaa Emara Rabee, Mebarek Lamara.

**Writing – original draft:** Amr A. Sayed Alahl, Mebarek Lamara, Suzanne L. Ishaq.

**Writing – review & editing:** Alaa Emara Rabee, Amr A. Sayed Alahl, Mebarek Lamara, Suzanne L. Ishaq.

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
