## [Decision Letter · Decision Letter 0]

12 Nov 2021

PONE-D-21-24385Fibrolytic rumen bacteria of camel and sheep and their applications in biofuel productionPLOS ONE

Dear Dr. Rabee,

Thank you for submitting your manuscript to PLOS ONE. After careful consideration, we feel that it has merit but does not fully meet PLOS ONE’s publication criteria as it currently stands. Therefore, we invite you to submit a revised version of the manuscript that addresses the points raised during the review process.

Please submit your revised manuscript within 30 days. If you will need more time than this to complete your revisions, please reply to this message or contact the journal office at plosone@plos.org. Please include the following items when submitting your revised manuscript:A rebuttal letter that responds to each point raised by the academic editor and reviewer(s). You should upload this letter as a separate file labeled 'Response to Reviewers'.A marked-up copy of your manuscript that highlights changes made to the original version. You should upload this as a separate file labeled 'Revised Manuscript with Track Changes'.An unmarked version of your revised paper without tracked changes. You should upload this as a separate file labeled 'Manuscript'.

We look forward to receiving your revised manuscript.

Kind regards,

Sabir Hussain

Academic Editor

PLOS ONE

Journal Requirements:

2. We note that you are reporting an analysis of a microarray, next-generation sequencing, or deep sequencing data set. PLOS requires that authors comply with field-specific standards for preparation, recording, and deposition of data in repositories appropriate to their field. Please upload these data to a stable, public repository (such as ArrayExpress, Gene Expression Omnibus (GEO), DNA Data Bank of Japan (DDBJ), NCBI GenBank, NCBI Sequence Read Archive, or EMBL Nucleotide Sequence Database (ENA)). In your revised cover letter, please provide the relevant accession numbers that may be used to access these data. For a full list of recommended repositories, see http://journals.plos.org/plosone/s/data-availability#loc-omics or http://journals.plos.org/plosone/s/data-availability#loc-sequencing.

This project was supported by the USDA National Institute of Food and Agriculture, Hatch Project Number ME0-022102 through the Maine Agricultural & Forest Experiment Station. Maine Agricultural and Forest Experiment Station Publication Number XXX.

Reviewers' comments:

Reviewer's Responses to Questions

**Comments to the Author**

1. Is the manuscript technically sound, and do the data support the conclusions?

Reviewer #1: Yes

2. Has the statistical analysis been performed appropriately and rigorously? 

Reviewer #1: Yes

3. Have the authors made all data underlying the findings in their manuscript fully available?

Reviewer #1: Yes

4. Is the manuscript presented in an intelligible fashion and written in standard English?

Reviewer #1: Yes

5. Review Comments to the Author

Reviewer #1: I think the research is interesting. However, was there anything novel about the functionality of the rumen microbial community in sheep and camels? was not sufficiently appealing to me. For example, it would be good if the differences between the rumen microbial community of sheep and camels and that of other rumen microbes could be more clearly expressed in graphs.

6. PLOS authors have the option to publish the peer review history of their article (what does this mean?). If published, this will include your full peer review and any attached files.

Reviewer #1: No

---

## [Author Response · Author response to Decision Letter 0]

29 Nov 2021

PlOS ONE editorial office

Fibrolytic rumen bacteria of camel and sheep and their applications in the bioconversion of barley straw to soluble sugars for biofuel production

We thank you and the Reviewers for their constructive comments and corrections. We responded to all comments and have extensively modified many sections in the paper to correct grammatical and style errors. I also enclosed unclean paper including all comments colored by yellow. Below are the responses to all the comments. We made a slight modification in the title of the manuscript. Our paper was improved greatly after the response to the reviewers’ comments.

We appreciate the opportunity to submit our manuscript to PlOS ONE.

Yours sincerely,

Dr. Alaa Rabee

Researcher at Desert Research Center, Egypt

Comments

Manuscript title: “Fibrolytic rumen bacteria of camel and sheep and their applications in the bioconversion of barley straw to soluble sugars for biofuel production”

In the unclean or marked manuscript you could notice that colored comments using yellow.

Responses

I think the research is interesting. However, was there anything novel about the functionality of the rumen microbial community in sheep and camels? was not sufficiently appealing to me. For example, it would be good if the differences between the rumen microbial community of sheep and camels and that of other rumen microbes could be more clearly expressed in graphs.

>> Thank so much, we added a small comparison at the end of discussion and added supplementary figure that compare the relative abundances of main bacterial groups in our study and in the rumen of other ruminant animals.

Figure1 Electron micrographs were not clear, so it is difficult to see the microorganisms attached to the surface of straws. Change to a clearer photo? Or isn't it necessary to include a photo?

>>We removed it, unfortunately we don’t have another photos. We had to take photos at different incubation times before 72 h to get clear photos for bacterial colonization.

Table1 Please add the unit of gas production.

>> Thank you, we added it and scanned all tables for similar mistaks.

Table1 "Inoculum Source" means “samples after 72 hours incubation”, is it correct? Please have a representation of what the sample is so that everyone can understand what it was.

>> Thank you, inoculum Source refers to animal source of rumen samples that were inoculated into barley straw plus bacterial media, camel or sheep; we clarified that throughout the manuscript.

Table 1　What does "overall" in Table 1 mean? 

I think that DNA extraction is extracting the solid (i.e., attached to the straw), but does "overall" include not only the solid but also the cultured liquid? Is it correct to say that the term "overall" includes not only solids but also cultured liquids? Please describe this in an easy-to-understand manner.

>> Thank you for this valuable comment. Overall refers to overall mean of the values in camels and sheep (6 samples: 3 for sheep and 3 for camels), we clarified that in all tables.

I did not understand, why you wrote camel rumen sample to “FC”, and inoculum sheep was “FR”. Wouldn't it be better to be consistent all the time in the paper, with camel samples and sheep samples?

>> Thank you, we modified that in the text and tables. We stuck with inoculum instead of sample throughout the manuscript, we modified that.

Tables 3 please add the units for value. (%).

>> We added it and scanned the tables for similar mistakes.

Line 435 : This time, the barley straw is autoclaved at 120 degrees Celsius, which I think is quite a heat treatment to the straw. This is a considerable heat treatment. It is possible that the effect of this heat treatment may have increased the hydrolysis efficiency, but what do you think about this? Also, I think this will be a problem when discussing the degradation efficiency when comparing with other papers. So, I think it would be better to have the results of how it was when it was not autoclaved.

>>> Thank you for this valuable comment. We used three serum bottles (medium + barley straw) without inoculation as blanks to allow the correction for dry matter disappearance during sterilization. However, I support your view; the microbial hydrolysis efficiency could be increased compared to the results of other studies. Therefore, we modified that section.

---

## [Editor Report · Decision Letter 1]

21 Dec 2021

Fibrolytic rumen bacteria of camel and sheep and their applications in the bioconversion of barley straw to soluble sugars for biofuel production

PONE-D-21-24385R1

Dear Dr. Rabee,

We’re pleased to inform you that your manuscript has been judged scientifically suitable for publication and will be formally accepted for publication once it meets all outstanding technical requirements.

Kind regards,

Sabir Hussain

Academic Editor

PLOS ONE
---

## [Editor Report · Acceptance letter]

27 Dec 2021

PONE-D-21-24385R1 

Fibrolytic rumen bacteria of camel and sheep and their applications in the bioconversion of barley straw to soluble sugars for biofuel production 

Dear Dr. Rabee:

I'm pleased to inform you that your manuscript has been deemed suitable for publication in PLOS ONE. Congratulations! Your manuscript is now with our production department. 

Kind regards, 

on behalf of

Dr. Sabir Hussain 

Academic Editor

PLOS ONE